

# Spin polarization induced by decoherence in a tunneling one-dimensional Rashba model

**Solmar Varela[1]** , **Mayra Peralta[1]** , **Vladimiro Mujica[2]** ,
**Bertrand Berche[3]** and **Ernesto Medina[4⋆]**

**1** Institute of Materials Science and Max Bergmann Center of Biomaterials,
Dresden University of Technology, 01062 Dresden, Germany
**2** School of Molecular Sciences, Arizona State University, Tempe, AZ 85281, United States
**3** Laboratoire de Physique et Chimie Théoriques, Université de Lorraine, Nancy, France
**4** Departamento de Física, Colegio de Ciencias e Ingeniería, Universidad San Francisco de Quito, Diego de Robles y Vía Interoceánica, Quito, 170901, Ecuador

⋆ emedina@usfq.edu.ec

## Abstract

Basic questions on the nature of spin polarization in two terminal systems and the way in which decoherence breaks Time-Reversal Symmetry (TRS) are analyzed. We exactly solve several one-dimensional models of tunneling electrons and show the interplay of spin precession and decay of the wavefunction in either a $U(1)$ magnetic field or an effective Spin-Orbit (SO) magnetic field. Spin polarization is clearly identified as the emergence of a spin component parallel to either magnetic field. We show that Onsager's reciprocity is fulfilled when time reversal symmetry is present and no spin polarization arises, no matter the barrier parameters or the SO strength. Introducing a Büttiker's decoherence probe, that preserves unitarity of time evolution, we show that breaking of TRS results in a strong spin polarization for realistic SO, and barrier strengths. We discuss the significance of these results as a very general scenario for the onset of the Chiral-Induced Spin Selectivity effect (CISS).



# 1  Introduction

The Spin-Orbit (SO) coupling is many times neglected in electron transport because of the energy scale of the coupling, meV for C, N, O in chiral molecules and e.g. Si, Ga, and Ge semiconductors in the bulk. Although the source of SO coupling in many technologically relevant materials is atomic, how this coupling translates to transport depends on the geometry of the connection between spin active atoms. This way in flat graphene, while atomic SO coupling is meV, the effective transport SO coupling is $\mu$eV. In a tight-binding model the SO coupling cancels from the nearest neighbour contribution due to interference effects and it is only when the second neighbours coupling is introduced one gets a meV interaction. On the other hand, bending the graphene sheet and producing e.g. nanotubes, increases the SO coupling by three orders of magnitude, as it becomes a first neighbours interaction [1]. The same enhancement is seen in silicene which has a corrugated surface structure that breaks the orthogonality of $\pi$ orbitals and the $\sigma$ structure in graphene [2].

In the case of electron transport in molecules, the SO coupling has been largely disregarded but has come into light due to large spin activity reported as Chiral-Induced Spin Selectivity (CISS) effect [3]. CISS effect is observed for both point chiral [4] in amino acids, helical chirality such as DNA [5], and helicene [6,7]. Biological molecules generally combine both as in e.g. oligopeptides [8,9].

There exists an enormous gap in understanding between the size of the SO coupling in molecules, in the meV range, and the magnitude of the spin polarization effect in CISS effect experiments, with spin-polarized transmissions above 40% [5]. This percentage exceeds the polarization strength produced by transmission through ferromagnets. The SO coupling is almost universally regarded as the spin active ingredient in CISS effect and theoretical estimates have yielded the correct qualitative behavior i.e. helicity states with the propagation axis as the quantization axis of the electron spin. On the other hand, the prediction for the magnitude of the spin polarization is at least ten times smaller, when correct atomic SO coupling strengths are contemplated [10–13].

An important issue on the symmetries involved in electron transmission with two terminals with SO coupling was pointed out by Yang, van der Wal and van Wees [14,15]. As was clearly argued, Onsager's reciprocity precludes the possibility of spin polarization in the two terminal setting in the linear regime, in contrast with the results of many works in the literature, both experimentally and numerically. It is then important to observe how symmetry arguments play out in specific calculations as reference results [14,16]. Symmetry arguments alone cannot say how sensitive these results will be in the face of weak symmetry-breaking perturbations, in this case, of Time-Reversal Symmetry (TRS).

In this work we will discuss the simplest transmission model through a SO active barrier, as tunneling is a very common electron transfer mechanism in large molecules [17]. We will explore the possibilities of spin-polarized electron transmission in a one-dimensional two-probe

setting for an exactly solved model. The action of a $U(1)$ field on tunneling electrons [18] will be contrasted with the effective momentum-dependent magnetic field arising from the SO coupling. A very important issue in the latter case is the velocity operator's non-diagonal nature that secures the flux's continuity through boundaries [19]. The strong result is that while a $U(1)$ magnetic field polarizes spin along its direction, under the action of the barrier, no spin polarization results under the action of a spin-orbit magnetic field. Following, we solve the model for the spin-orbit magnetic field under the effects of weak TRS breaking decoherence effects, enacted through Büttiker's probe. Large spin polarization highlights a high sensitivity to TRS breaking with realistic SO couplings. These findings address the core issue of CISS effect.

To clarify the nature of the contribution in this work: The first three sections are qualitative, as the parameters are not chosen to fit the actual physical ones for chiral molecules. In section 3, independently of the parameters chosen, the null results for polarization are exact results verifying the symmetry requirements of the Onsager relations. A few previous numerical computations resulted in transport spin filtering associated with non-hermiticities that crept into the calculations. Although we refer to previous papers with the correct conclusion for the polarization of the Rashba case, these contributions have shortcomings that are overcome here realizing the correct boundary conditions.

The meaning of polarization in terms of an asymmetric treatment of opposite spin orientations is also clarified in this paper, along with an exact solution to the problem (with time-reversal symmetry in place), in light of Buttiker's magnetic field case that breaks time reversal. Finally, in section 5, on Buttiker's probe, the parameters are fitted to a polaron model of transport. Nevertheless, the coupling to the reservoir is not parameterized quantitatively and we only show qualitatively the sensitivity of the polarization to decoherence. We also note that in more realistic situation, the Rashba coupling can be a varying function of the coordinate [20,21], since it can be modulated by local geometrical configurations [22], but we believe that our findings where this is neglected retain the essential of the physics at work..

## 2 Barrier model with a magnetic field

### 2.1 Spectrum, eigenfunctions, and wavevectors

We will first fully solve analytically for the emblematic problem of a magnetic field under a barrier for spinful particles [18]. The correct solution to this problem allowed for properly addressing the tunneling time problem. Büttiker realised that spin precession in the field is modulated by the spin-dependent decay of the wavefunctions under the barrier, generating polarization in the direction of the magnetic field. The Hamiltonian in this case is given by

$$\mathcal{H} = \begin{cases} \left(\frac{p_x^2}{2m} + V_0\right)\mathbb{1}_\sigma - \Gamma\sigma_z\,, & \text{if } 0 < x < a\,, \\ \left(\frac{p_x^2}{2m}\right)\mathbb{1}_\sigma\,, & \text{otherwise}\,, \end{cases} \tag{1}$$

where $\mathbb{1}_\sigma$ is the unit matrix in spin space and $\sigma_i$ are the Pauli spin matrices, with $i = x, y, z$. $\Gamma = \hbar\omega_L/2$ where $\omega_L$ is the Larmor frequency, and $V_0$ is the barrier height. $\mathcal{H}$ acts on the spinors $\psi = (\psi_+(x)\ \psi_-(x))$ where $|\psi_\pm|^2 dx$ is the probability of find a particle between $x$ and $x + dx$ with spin $\pm\hbar/2$. The Hamiltonian inside the barrier has the dispersion relation depicted in Fig. 1. The choice of coordinates is slightly different from that of Büttiker so we can discuss all one-dimensional models with the same notation.

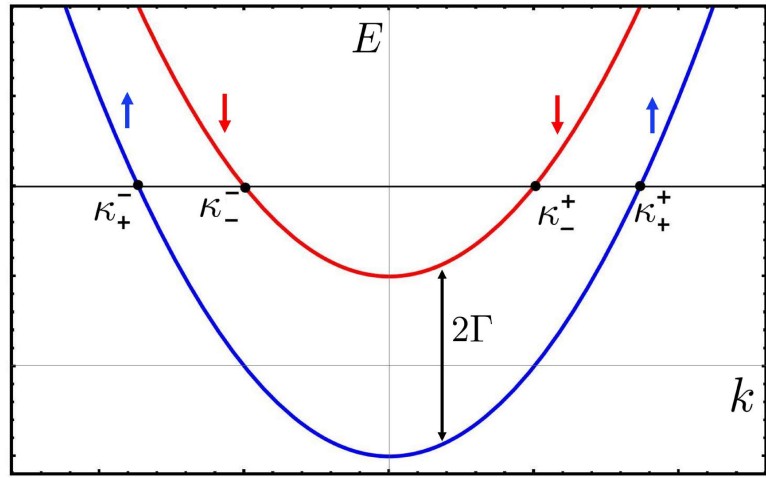

Figure 1: The dispersion relation for the barrier with a magnetic field. The figure depicts the degenerate $\kappa_\sigma^\lambda$ vectors that occur in the barrier range.

The incoming wavefunction we choose to be

$$\psi = \frac{1}{\sqrt{1+|s|^2}} \begin{pmatrix} 1 \\ is \end{pmatrix}. \tag{2}$$

The values of $s = \pm 1$ correspond to the two eigenfunctions of the $\sigma_y$ matrix, and $s = \pm i$ correspond to the two eigenfunctions of the $\sigma_x$ matrix, appropriately normalized. The eigenvalues of the Hamiltonian are

$$E = \frac{p_x^2}{2m} - \sigma\Gamma + V_0 \,, \tag{3}$$

where $\sigma = \pm 1$ is the spin degree of freedom. Using $E = \hbar^2 k^2/2m$, we define the wavevector outside the barrier $k$. This Hamiltonian is not time-reversal invariant since inverting time flips $p_x$ and $\sigma$, and these flips change the energy. From the eigenvalue equation, one can then distinguish between the different wavevectors under the barrier

$$\kappa_\sigma^\lambda = \lambda \left( k^2 - k_0^2 + \sigma k_B^2 \right)^{1/2} \,, \tag{4}$$

where $k_B^2 = 2m\Gamma/\hbar^2$ and $k_0^2 = 2mV_0/\hbar^2$. As can be seen, $\kappa_\sigma^\lambda$ can only be real or imaginary. Thus we have either exponentially decaying solutions for $k^2 < k_0^2 - \sigma k_B^2$ or plane waves otherwise. As the Hamiltonian commutes with $\sigma_z$ in the barrier region we can superpose eigenfunctions of $\sigma_z$ as

$$\psi_1 = \frac{1}{\sqrt{1+|s|^2}} \begin{pmatrix} 1 \\ is \end{pmatrix} e^{ikx} + \begin{pmatrix} A_+ \\ A_- \end{pmatrix} e^{-ikx}, \tag{5}$$

$$\psi_2 = \epsilon \begin{pmatrix} 1 \\ 0 \end{pmatrix} e^{i\kappa_+^+ x} + \zeta \begin{pmatrix} 0 \\ 1 \end{pmatrix} e^{i\kappa_-^+ x} + \eta \begin{pmatrix} 1 \\ 0 \end{pmatrix} e^{i\kappa_+^- x} + \theta \begin{pmatrix} 0 \\ 1 \end{pmatrix} e^{i\kappa_-^- x}, \tag{6}$$

$$\psi_3 = \begin{pmatrix} D_+ \\ D_- \end{pmatrix} e^{ikx} \,. \tag{7}$$

The boundary conditions are

$$\psi_i(x_b) = \psi_{i+1}(x_b), \tag{8}$$

$$\hat{v}_x \psi_i(x_b) = \hat{v}_x \psi_{i+1}(x_b), \tag{9}$$

where $\hat{v}_x = (\partial \mathcal{H}/\partial p_x)$ is the velocity operator and $x_b$ the boundary between the different space regions. We match the wavefunction and the amplitude flux. This latter boundary condition is very important to realize and has often been confused in the literature. Any dependence of the mass on position (effective mass) should be carefully considered to yield the appropriate hermitian velocity operator [23]. If the mass is a constant, then the boundary conditions amount to matching the wavefunctions and the first derivatives thereof. Although this problem can be separated into two spinless tunneling problems with different barrier heights [18], we have decided to phrase somewhat more cumbersomely to make a few important points when considering spin-orbit coupling.

The system of equations above can be solved to yield

$$
\begin{aligned}
D_+ &= \frac{t_+}{\sqrt{1+|s|^2}} = \frac{2k(\kappa_+^- - \kappa_+^+)e^{-ia(-\kappa_+^- - \kappa_+^+ + k)}}{\sqrt{|s|^2+1}\left(e^{ia\kappa_+^-}(\kappa_+^- - k)(\kappa_+^+ + k) - e^{ia\kappa_+^+}(\kappa_+^- + k)(\kappa_+^+ - k)\right)}, \\
D_- &= \frac{ist_-}{\sqrt{1+|s|^2}} = \frac{2iks(\kappa_-^- - \kappa_-^+)e^{-ia(-\kappa_-^- - \kappa_-^+ + k)}}{\sqrt{|s|^2+1}\left(e^{ia\kappa_-^-}(\kappa_-^- - k)(\kappa_-^+ + k) - e^{ia\kappa_-^+}(\kappa_-^- + k)(\kappa_-^+ - k)\right)}, \\
A_+ &= \frac{r_+}{\sqrt{1+|s|^2}} = \frac{(\kappa_+^- - k)(k - \kappa_+^+)\left(e^{ia\kappa_+^-} - e^{ia\kappa_+^+}\right)}{\sqrt{|s|^2+1}\left(e^{ia\kappa_+^-}(\kappa_+^- - k)(\kappa_+^+ + k) - e^{ia\kappa_+^+}(\kappa_+^- + k)(\kappa_+^+ - k)\right)}, \\
A_- &= \frac{isr_-}{\sqrt{1+|s|^2}} = \frac{is(\kappa_-^- - k)(k - \kappa_-^+)\left(e^{ia\kappa_-^-} - e^{ia\kappa_-^+}\right)}{\sqrt{|s|^2+1}\left(e^{ia\kappa_-^-}(\kappa_-^- - k)(\kappa_-^+ + k) - e^{ia\kappa_-^+}(\kappa_-^- + k)(\kappa_-^+ - k)\right)},
\end{aligned}
\tag{10}
$$

where the $t_\pm$ and $r_\pm$ denote the transmission and reflection amplitudes. We recall $\kappa_\sigma^\lambda = \lambda\sqrt{\left(k^2 - k_0^2 + \sigma k_B^2\right)}$. In the next section, we obtain the behavior of the spin as a function of the magnetic field strength consistent with tunneling and we see both regular Larmor precession with $V_0 = 0$, and precession combined with spin alignment in the field direction when tunneling occurs.

## 2.2 Spin precession under the barrier with magnetic field

One readily verifies the differential decay of the transmission with the length of the barrier as $T_+ \sim e^{-2\kappa_+^+ a}$, and $T_- \sim e^{-2\kappa_-^+ a}$ as long as $E < V_0 \mp \hbar\omega_L/2$. The following relations quantify the polarization of the electron, the transmitted (T) wave is

$$
\psi_T = \frac{1}{\sqrt{|D_+|^2 + |D_-|^2}}\begin{pmatrix} D_+ \\ D_- \end{pmatrix}e^{ikx},
\tag{11}
$$

and the spin averages are defined by

$$
\langle s_z \rangle = \frac{\hbar}{2}\langle \psi_T | \sigma_z | \psi_T \rangle = \frac{\hbar}{2}\frac{|D_+|^2 - |D_-|^2}{|D_+|^2 + |D_-|^2},
$$

$$
\langle s_y \rangle = \frac{\hbar}{2}\langle \psi_T | \sigma_y | \psi_T \rangle = i\frac{\hbar}{2}\frac{D_+ D_-^* - D_+^* D_-}{|D_+|^2 + |D_-|^2},
$$

$$
\langle s_x \rangle = \frac{\hbar}{2}\langle \psi_T | \sigma_x | \psi_T \rangle = \frac{\hbar}{2}\frac{D_+ D_-^* + D_+^* D_-}{|D_+|^2 + |D_-|^2}.
\tag{12}
$$

Analogous relations can be written for the reflected wave. A spin oriented in the $y$ direction (corresponding to $s = -1$, see Eq. (2) will Larmor precess around the magnetic field (in $z$ direction) when $V_0 = 0$ as shown in Fig. 2.

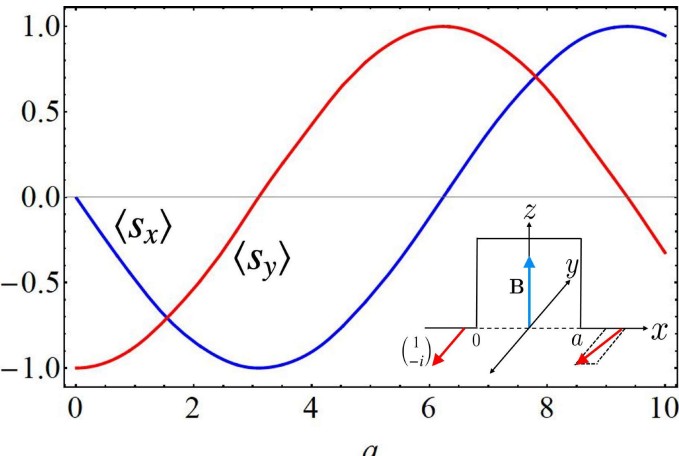

Figure 2: Precession of the spin as it goes through an increasing barrier length $a$ starting from the $1/\sqrt{2}(1 \quad -i)$ state, for $V_0 = 0$. This is the simple Larmor precession initially surmised for tunneling times [18].

On the other hand, for $V_0 > E \pm \hbar\omega_L/2$, spin precession around the magnetic field is only part of the average spin motion, since each spin component decays at a different rate under the barrier. This gives rise to a $z$-component that aligns with the direction of the field [18]. Figure 3 depicts the qualitative motion for the latter case. For $V_0 < E \pm \hbar\omega_L/2$ only Larmor precession follows.

Finally, Fig. 4 shows the transmitted probability in $z$ quantization axis. The input spin orientation is along the negative $y$ axis and the transmitted wave selects the up spin orientation due to the slower decay of the lower energy state under the barrier. This produces spin alignment with the magnetic field.

A very important relationship to check is that of the conservation of angular momentum. As proven in reference [18], the conservation can be stated exactly as

$$(R_+ + R_-)\langle s_z \rangle_R = -\langle s_z \rangle (T_+ + T_-), \tag{13}$$

where $T_+ = |t_+|^2$ and $R_+ = |r_+|^2$ and $\langle s_z \rangle_R$ is the reflected (R) spin component in the $z$ direction.

Such a relationship is verified in Fig. 5 where the changed angular momentum transmitted is compensated by the opposite angular momentum reflected. We have thus verified Büttiker's scenario for tunneling with a magnetic field under the barrier. Before addressing the case of the SO coupling in one dimension, some useful gauge concepts will be introduced.

# 3 Barrier model with a Rashba term

## 3.1 Spectrum, eigenfunctions, and wavevectors

We solve the scattering problem for the following model

$$\mathcal{H} = \begin{cases} (\frac{p_x^2}{2m} + V_0)\mathbb{1}_\sigma + \Lambda p_x \sigma_y, & \text{if } 0 < x < a, \\ (\frac{p_x^2}{2m})\mathbb{1}_\sigma, & \text{otherwise}, \end{cases} \tag{14}$$

where $\mathbb{1}_\sigma$ is the unit matrix in spin space and $\sigma_i$ are the Pauli spin matrices. This Hamiltonian can be obtained from a helical model of a molecule with $p$ wave overlaps and SO active car-

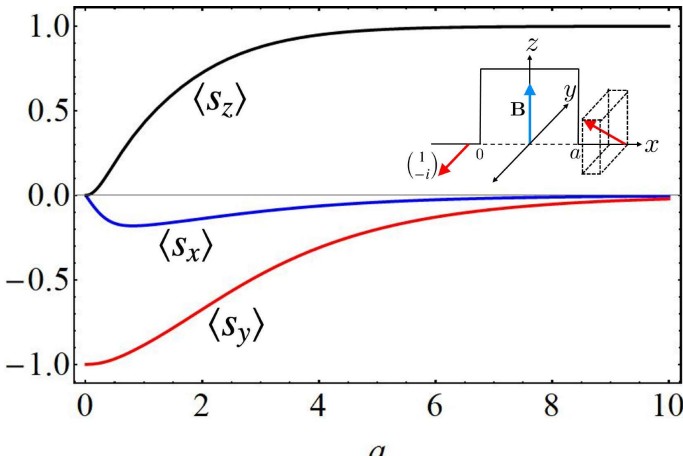

Figure 3: Relaxation of spin toward magnetic field direction due to tunneling when $k^2 < k_0^2 \mp k_B^2$ [18]. For long enough barriers the spin becomes completely polarized in the direction of the field. The reference values taken for the plot are $k = 2/a$, $k_0 = 3/a$, and $k_B = 1/a$, where $a$ is the barrier length.

bon/nitrogen atoms [24–26]. The magnitude of the SO coupling considered is in fact derived from the overlaps that are only feasible for the chiral structure considered in those models.

We take the incident beam to have an amplitude

$$\psi = \frac{\sqrt{2}}{\sqrt{1+s^2}} \begin{pmatrix} \frac{1+s}{2} \\ \frac{1-s}{2} \end{pmatrix}, \tag{15}$$

where $s = 1$ corresponds to the up-spin normalized eigenstate of the $\sigma_z$ matrix and $s = -1$ to the down-spin state. The normalization also allows access to all spin states in the $x-z$ plane of the Bloch sphere. Here we will illustrate how the $SU(2)$ gauge vector for the one-dimensional Rashba Hamiltonian becomes crucial in the barrier boundary conditions [19] which has been missed in previous treatments.

We can faithfully rewrite the Hamiltonian in the following form

$$\mathcal{H} = \frac{1}{2m} \left( \hat{p}_x \mathbb{1}_\sigma + m\Lambda(x)\sigma_y \right)^2 + V_0 - \frac{m\Lambda^2}{2}, \tag{16}$$

where we can identify the $SU(2)$ gauge field $\mathcal{A}_x = A_x^y \sigma_y = m\Lambda(x)\sigma_y$. The velocity operator defined by $v_x = \partial \mathcal{H}/\partial p_x = ((p_x/m)\mathbb{1}_\sigma + \Lambda\sigma_y)$, where no effective mass differences are considered [23] for the different scattering regions. Solving for the eigenvalues of this Hamiltonian, we arrive at

$$E = \frac{1}{2m}(p_x + m\sigma\Lambda)^2 - \frac{m\Lambda^2}{2} + V_0, \tag{17}$$

where $\sigma = \pm 1$ is the spin quantum number (eigenvalue label of $SU(2)$ Hamiltonian). Equating $E = \hbar^2 k^2/2m$ we define the wavevector outside the barrier region as $k$. Starting from the eigenvalue, we can solve for the possible values of $p_x = \hbar q$. A new quantum number arises that distinguishes right and left propagating waves. The resulting possible values of the wavevector under the barrier are

$$q_\sigma^\lambda = \lambda\sqrt{k^2 + k_{so}^2 - k_0^2} - \sigma k_{so}, \tag{18}$$

where $k_{so} = m\Lambda/\hbar$ and $k_0^2 = 2mV_0/\hbar^2$. The meaning of the quantum numbers is depicted in Fig. 6, where the degeneracy of two Kramer's pairs is evident. Note that for each direction of propagation, there are two distinct wavevectors with opposite spin labels and that the previous

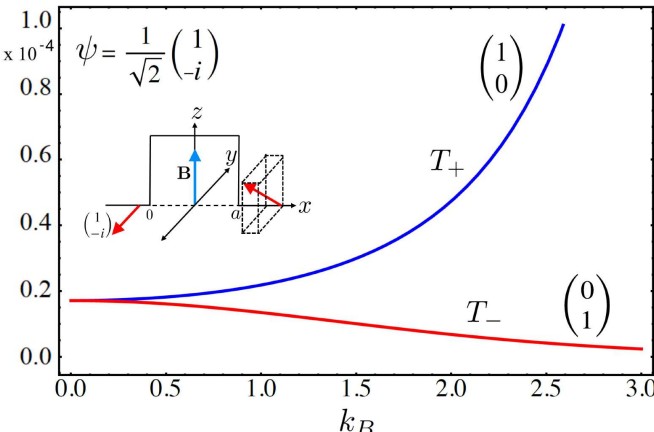

Figure 4: Transmission contrast between spin components for a magnetic field under the barrier. One can see the preferred spin polarization due to the slower decay of the lower energy spin configuration under the barrier, leading to an alignment of the entering spin to the magnetic field. The parameters chosen are the same as in Fig. 3.

wavevector can be real or *complex* depending on the values of the incoming wavevector (with energy $E = \hbar^2 k^2 / 2m$) and the height of the potential barrier. As is easily derived from Eq. (14), the Hamiltonian commutes with $\sigma_y$, and $\hat{p}_x$ so it has common eigenstates with $\sigma_y$ and the $\hat{p}_x$ eigenstates. In the $\sigma_z$ basis the wavefunctions in the different regions are parameterized as follows

$$
\psi_1 = \begin{pmatrix} \frac{1+s}{2} \\ \frac{1-s}{2} \end{pmatrix} e^{ikx} + \begin{pmatrix} A_+ \\ A_- \end{pmatrix} e^{-ikx},
$$

$$
\psi_2 = \frac{\alpha}{\sqrt{2}} \begin{pmatrix} 1 \\ i \end{pmatrix} e^{iq_+^+ x} + \frac{\beta}{\sqrt{2}} \begin{pmatrix} 1 \\ -i \end{pmatrix} e^{iq_-^+ x} + \frac{\gamma}{\sqrt{2}} \begin{pmatrix} 1 \\ i \end{pmatrix} e^{iq_+^- x} + \frac{\delta}{\sqrt{2}} \begin{pmatrix} 1 \\ -i \end{pmatrix} e^{iq_-^- x},
$$

$$
\psi_3 = \begin{pmatrix} D_+ \\ D_- \end{pmatrix} e^{ikx}, \tag{19}
$$

where the coupling between the direction of propagation and spin orientation has been implemented by the appropriate $q_\sigma^\lambda$ wavevectors. The boundary conditions are the same as in Eqs.9 [19] where $\hat{v}_x = (\hat{p}_x + m \Lambda \sigma_y)/m$. The linear system of eight unknowns can be explicitly solved for the transmission and reflection amplitudes

$$
t_+ = \frac{(1+i)\Delta k e^{-iak}\left((1-is)e^{ik_{so}a} + (s-1)e^{-ik_{so}a}\right)}{(e^{-ia\Delta}(\Delta+k)^2 - e^{ia\Delta}(k-\Delta)^2)},
$$

$$
t_- = -\frac{(1+i)\Delta k e^{-iak}\left((s+i)e^{ik_{so}a} - (1+is)e^{-ik_{so}a}\right)}{(e^{-ia\Delta}(\Delta+k)^2 - e^{ia\Delta}(k-\Delta)^2)},
$$

$$
r_+ = \frac{(k-\Delta)(\Delta+k)(s+1)\left((k-\Delta)^2 e^{2ia\Delta} + (\Delta+k)^2 e^{-2ia\Delta} - 2(k^2+\Delta^2)\right)}{2\left(e^{-ia\Delta}(\Delta+k)^2 - e^{ia\Delta}(k-\Delta)^2\right)^2},
$$

$$
r_- = -\frac{(k-\Delta)(\Delta+k)(s-1)\left((k-\Delta)^2 e^{2ia\Delta} + (\Delta+k)^2 e^{-2ia\Delta} - 2\left(k^2+\Delta^2\right)\right)}{2\left(e^{-ia\Delta}(\Delta+k)^2 - e^{ia\Delta}(k-\Delta)^2\right)^2}, \tag{20}
$$

where $\Delta = \sqrt{k^2 + k_{so}^2 - k_0^2}$, and as before $t_\pm$ and $r_\pm$ are the spin dependent transmission and reflection amplitudes. Such amplitudes will be very important to understand how decoherence

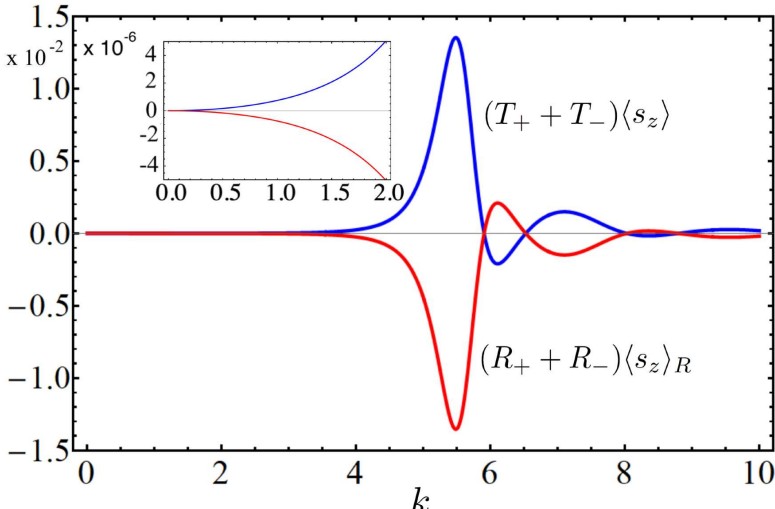

Figure 5: Angular momentum conservation balancing transmitted spin up ($z$-component) and reflected spin down. As there is no incident spin-up current the two previous components (Eq. (13)) must balance.

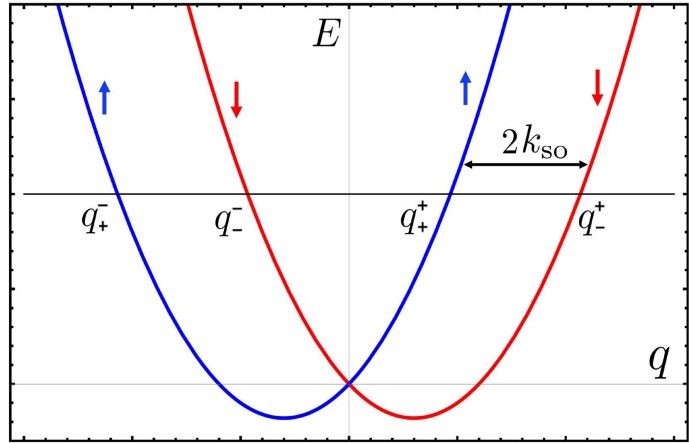

Figure 6: Dispersion for Hamiltonian in Eq. (14) The labels correspond to the wavevectors in the barrier region with $q_\sigma^\lambda$.

effects generate spin polarization. The barrier region amplitudes are

$$
\alpha = \frac{(-1)^{1/4} e^{iaq_+^-} k(s-i)(k+\Delta)}{-e^{iaq_+^+}(k-\Delta)^2 + e^{iaq_+^-}(k+\Delta)^2},
$$
$$
\beta = \frac{(-1)^{1/4} k(1-is)e^{iaq_-^-}(\Delta+k)}{e^{iaq_-^-}(\Delta+k)^2 - e^{iaq_-^+}(k-\Delta)^2},
\tag{21}
$$

and

$$
\gamma = \frac{(-1)^{1/4} k(s-i)e^{iaq_+^+}(k-\Delta)}{e^{iaq_+^+}(k-\Delta)^2 - e^{iaq_+^-}(\Delta+k)^2},
$$
$$
\delta = \frac{(-1)^{1/4} k(1-is)e^{iaq_-^+}(k-\Delta)}{e^{iaq_-^+}(k-\Delta)^2 - e^{iaq_-^-}(\Delta+k)^2}.
\tag{22}
$$

We recall that $q_\sigma^\lambda = \lambda\sqrt{k^2 + k_{so}^2 - k_0^2} - \sigma k_{so}$.

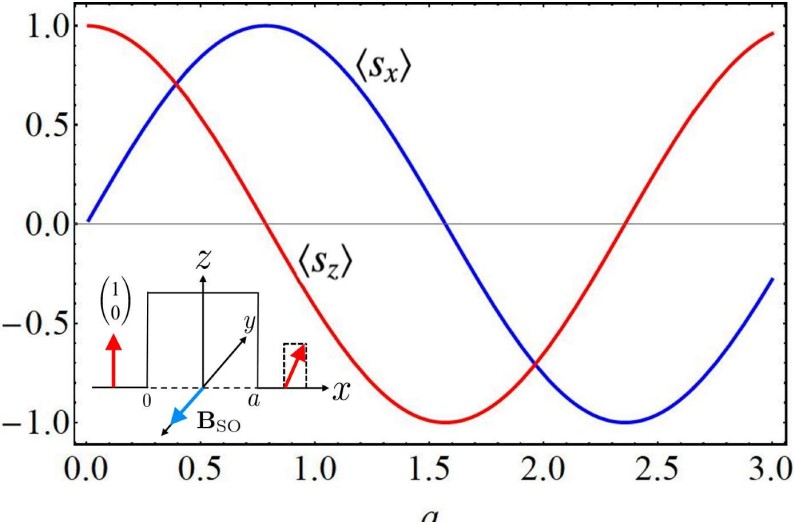

Figure 7: Precession of spin around the spin-orbit magnetic field. Note that as the $k$ vector inside the barrier has always a real part, we have pure precession with no tilting toward the magnetic field as with the magnetic field in the previous section. This happens below and above the barrier.

### 3.2 Spin precession under the barrier for Rashba

The transmission of up-spin as a function of the entry spin polarization is

$$
\begin{aligned}
|t_+|^2 &= \frac{8|\Delta|^2 k^2[1 + s\cos(2k_{\text{so}}a)]}{|e^{-ia\Delta}(\Delta + k)^2 - e^{ia\Delta}(k-\Delta)^2|^2}, \\
|t_-|^2 &= \frac{8|\Delta|^2 k^2[1 - s\cos(2k_{\text{so}}a)]}{|e^{-ia\Delta}(\Delta + k)^2 - e^{ia\Delta}(k-\Delta)^2|^2},
\end{aligned}
\tag{23}
$$

from where we can see that $T = |t_+|^2 + |t_-|^2$ so the total conductance is

$$
\begin{aligned}
G_{\text{total}} &= G_+ + G_- \\
&= \frac{e^2}{h}T = \frac{16e^2|\Delta|^2 k^2}{h\,|e^{-ia\Delta}(\Delta + k)^2 - e^{ia\Delta}(k-\Delta)^2|^2},
\end{aligned}
\tag{24}
$$

where $G_\pm = (e^2/h)|t_\pm|^2$, which is spin independent [19,27] even in the presence of a barrier and an open system at either end of the barrier.

Following the definitions for average spin components in Eq. (.12), we can now see the behavior of the injected spin into the spin-orbit active barrier. Fig. 7 shows how the average spin traverses the spin-active barrier region with SO coupling. In analogy with Büttiker's $U(1)$ magnetic field, we can define a new momentum-dependent magnetic field $\mathbf{B}_{\text{SO}}$ given the mapping $\lambda p_x \sigma_y = -\gamma \mathbf{B}_{\text{SO}} \cdot \boldsymbol{\sigma}$ that results in $\mathbf{B}_{\text{SO}} = -(\Lambda/\gamma)p_x \mathbf{u}_y$. $\mathbf{B}_{\text{SO}}$ lies in the negative $y$ direction for the model Hamiltonian. Precession follows correctly the torque equation $d\langle\mathbf{s}\rangle/dt = \gamma\langle\mathbf{s}\rangle \times \mathbf{B}_{\text{SO}}$. Note that, as even under the barrier, the wavevector is complex, unlike the magnetic field case, precession proceeds with no generation of a spin component along the $\mathbf{B}_{\text{SO}}$ direction. Also, both spin components suffer the same decay within the barrier (although dependent on SO) independent of their spin orientation (see Eq. (18)).

We can see from comparing the two cases (magnetic field and SO) that one superposes different $k$ vectors corresponding with the same energy when traversing the spin active region.

The two wavevectors, having different real parts, cause precession due to a torque around the direction of $\mathbf{B}_{SO}$. In the case of a real magnetic field, under the barrier, the $k$ vector is purely imaginary, and only a spin-dependent decay ensues (see Eq. (4)), producing an alignment of the spin in the magnetic field direction. In the case of the SO, there is always a real part to the $k$ vector (even under the barrier) so that precession occurs for energies above and below the barrier. No spin polarization along $\mathbf{B}_{SO}$ follows from this scenario in tune with time-reversal symmetry.

## 4 Other helicity Hamiltonians

Varying the Hamiltonian under the barrier to the case where the eigenstates are projected along the direction of propagation (helicity states), can be interpreted directly from the previous results. The Hamiltonian in this case is

$$
\mathcal{H} = \begin{cases} \left(\frac{p_x^2}{2m} + V_0\right)\mathbb{1}_\sigma + \Lambda p_x \sigma_x, & \text{if } 0 < x < a, \\ \left(\frac{p_x^2}{2m}\right)\mathbb{1}_\sigma, & \text{otherwise.} \end{cases} \tag{25}
$$

The SO magnetic field is now in the $x$ direction as $\mathbf{B}_{SO} = -\Lambda p_x/\gamma \; \mathbf{u}_x$. Working out the eigenstates in the SO active region, the eigenstates will be those of $\sigma_x$ matrix, and the possible $k$ vectors will be $\kappa_\sigma^\lambda = \lambda\sqrt{k^2 + k_{so}^2 - k_0^2} - \sigma k_{so}$ as before. The $k$-vector under the barrier always has a real part that results in a spin precession. If we start from a spin orientation in the $z$-axis, then the spin will precess around the $x$ axis without generating a spin component in the $x$ direction. So no changes from the conclusion in the previous section follow in this case.

## 5 Decoherence with Büttiker's probe

The spin-orbit coupling does not contrast between spin species, so it cannot, alone, account for polarised spin polarization as expected in CISS effect. Nevertheless, the perfect conditions under which these results are valid i.e., no coupling to a TRS breaking probe beyond the two terminals, are not met, especially at room temperature conditions. A thermalization of electron transport to the environment through the electron-phonon or electron-electron interactions is inevitable. This environment can be modeled as a lumped probe that disrupts the delicate coherences that yield Bardarson's theorem that translates into transport as the Onsager reciprocity relations in the linear regime. This turns our attention to a tunneling molecular system to a three-probe scenario.

The Büttiker's voltage probe [28] is an ingenious way to introduce decoherence processes through the scattering matrix for an exactly solved model. Here we introduce a generalization of the probe used previously in the context of persistent currents [29–31] (see Fig. 8). The probe is spin insensitive, so we do not introduce extraneous sources of spin selection. This is achieved by introducing two probes, one for each spin species at the same point connected to a third reservoir thermalized to a Fermi distribution at temperature $T$. The probe is wide-band, supporting the wavevectors injected by the barrier channels. The probe scattering matrix returns an amplitude consistent with a simple electron reservoir unrelated to the input amplitude (while preserving unitarity) so that a disruption to the interferences occurs according to the local fluxes of each spin orientation. Such a probe introduces TRS breaking that generates the Büttiker tilting of the spin in the $\mathbf{B}_{SO}$ direction producing net spin polarization.

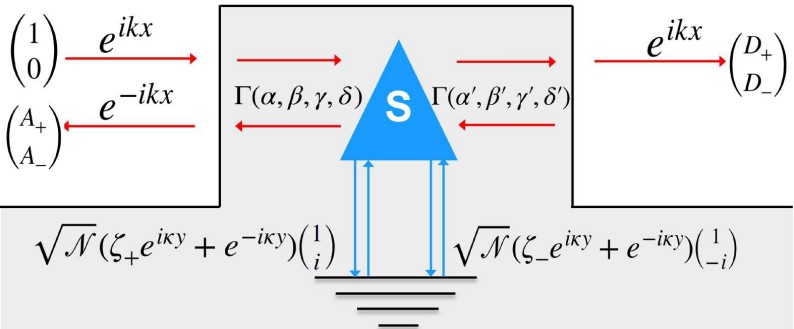

Figure 8: Büttiker's probe under the spin-orbit active barrier with electrons impinging on the left and leaving on the rigth with wavevector $k$. The function $\Gamma(\alpha, \beta, \gamma, \delta)$ represents the wavefunction combination under the barrier as in Eq. A.1. The probe absorbs each eigenstate spin species under the barrier with the same scattering matrix, so no spurious spin selection is induced. Flux conditions are imposed on building the S matrix for a wideband Büttiker's probe.

The behavior of the Büttiker's probe follows the combination of an ideal lead with $v = \hbar \kappa / m$ that supports a current $dI = ev(dN/dE)f(E)dE$ in the energy interval $dE$, where $f(E)$ is the Fermi distribution, $dN/dE = 1/2\pi\hbar v$ is the density of states. This model can then induce level broadening [29, 30], ( [31] for Hamiltonian version) and level shifts under the barrier, and also depends on where the decoherence event occurs. Besides the coupling of the probe to the barrier, we can also control the temperature through the Fermi distribution of the attached reservoir. The Büttiker's probe is also discussed in [32], where they consider inelastic events i.e., energy changes but conservation of particles. Here $S^\dagger S = 1$ so that only decoherence is contemplated.

Figure 8 shows the four regions that must be matched for continuity and flux. Under the barrier, the matching occurs at position $(x_0, y_0) = (x_0, 0)$ where $y$ describes the coordinate of the third probe. The Scattering ($S$) matrix can then emulate a generic dephasing process [29]. Matching flux conditions at $x_0$ yields the following $S$ matrix between input and output amplitudes for each spin species i.e., spin eigenstates under the barrier. Here we only show the spin down matrix equation (see Appendix A for details)

$$\Psi_{out} = \begin{pmatrix} \sqrt{\mathcal{N}}\zeta_- \\ \beta' \\ \delta \end{pmatrix} = S_- \Psi_{in} = \begin{pmatrix} -(\mathcal{A} + \mathcal{B}) & -\sqrt{\varepsilon}e^{iq_-^- x_0} & \sqrt{\varepsilon}e^{iq_-^+ x_0} \\ \sqrt{\varepsilon}e^{-iq_-^+ x_0} & -\mathcal{A}e^{-2i\Delta x_0} & \mathcal{B} \\ -\sqrt{\varepsilon}e^{-iq_-^- x_0} & \mathcal{B} & -\mathcal{A}e^{2i\Delta x_0} \end{pmatrix} \begin{pmatrix} \sqrt{\mathcal{N}} \\ \delta' \\ \beta \end{pmatrix}, \qquad (26)$$

where $\Psi_{in,out}$ represent the input/output amplitudes to the junction and $S_-$ is the scattering matrix for the spin-down label. The labels follow the usage previously introduced where $q_\sigma^\lambda$, with $\sigma$ the spin label and $\lambda$ the sense of propagation label. $\mathcal{A} = (\sqrt{1-2\varepsilon} - 1)/2$ and $\mathcal{B} = (\sqrt{1-2\varepsilon} + 1)/2$, while $\mathcal{N} = ef(E)dE/2\pi\hbar v$ with $f(E)$ the Fermi distribution, $e$ the electron charge, $E$ the energy and $v$ the velocity of the carriers in the lead [28]. $0 < \varepsilon < 0.5$ describes the coupling of the probe to the barrier from uncoupled to fully coupled.

The results regarding the influence of decoherence, barrier length, and SO coupling is depicted in Figs. 9 thru 11. In Fig. 9, where only qualitative parameters are used so that precession can be appreciated, one can see that the SO coupled to the third probe produces a smooth disruption of the spin precession which is no longer in the $(x, z)$ plane but yields a $\langle s \rangle_y$ component. The tunneling electrons now achieve a large polarization in the direction of the SO magnetic field (see inset). Thus, the $\mathbf{B}_{SO}$ acts as a symmetry-breaking interaction such as a real magnetic field in Fig. 3.

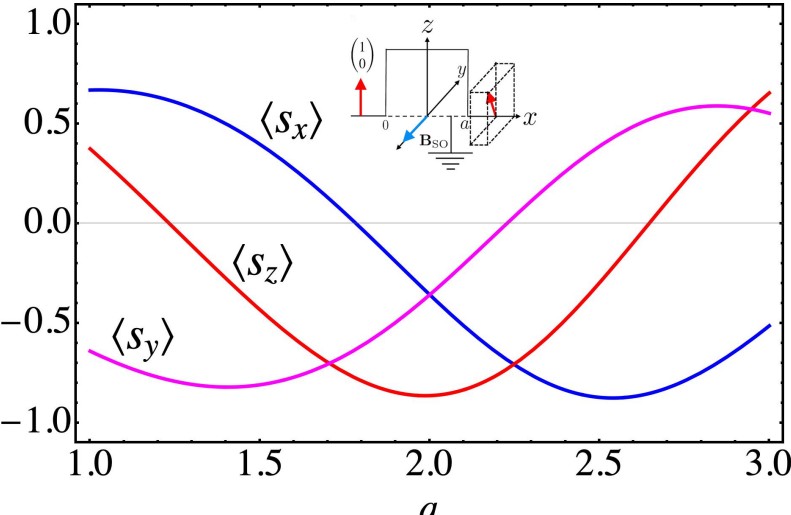

Figure 9: Precession of input spin orientation when the decoherence probe couples to a particular point $x_0$ under the barrier. A noticeable disruption of spin precession is observed, generating a spin polarization toward $B_{SO}$ analogous to an actual magnetic field (see Fig. 3, realizing broken time-reversal symmetry). As here $x_0 = 0.8$, we begin the plot at $a = 1$ so that the probe is always under the barrier.

Figure 10 depicts the transmission contrast between spin components along the $y$ axis in which the $\mathbf{B}_{SO}$ field is oriented. As can be seen, the spin acquires a preferred direction along the $-y$ direction aligning with the SO magnetic field. This is analogous to the case of the TRS breaking $U(1)$ magnetic field considered at the outset. When the SO strength is zero, the two components merge, showing no spin activity, and only express the transmission through the barrier.

Figure 11 shows the dependence of the new polarization as a function $k_{so}$, and the coupling to the reservoir. In contrast to previous figures depicting qualitative precession features, here we have set the parameters to realistic ranges for spin-orbit strength, barrier height, and barrier-probe coupling [24]. It is evident that even for weak coupling ($\varepsilon$) and $E_{SO} \sim 10$ meV, one can achieve polarizations of 40%. The polarization effects can yield positive and negative polarizations depending on the length of the barrier $a$ and exhibit a non-trivial temperature dependence. No spin polarization is produced without the SO interaction, no matter the coupling to the third probe.

Figure 12 shows the non-monotone/interference effects of coupling to the third probe and the sensitivity to the coupling to the third probe. Sufficiently large values of SO wavevector $k_{so}$, can produce a large polarization at low coupling, while large couplings degrade the polarization. Figure 13 depicts the temperature dependence of the polarization as a function of the barrier length. The temperature dependence is expressed through the parameter $\mathcal{N} = e f(E) dE / 2\pi \hbar \nu$ proportional to the thermal occupation of the probe. As temperature rises $\mathcal{N}$ is reduced so that as the temperature is increased, the polarization increases for fixed barrier length. Non-monotone effects with the probe coupling can change this temperature dependence. They will be determined by the specific nature of how the electron spin current thermalizes to the environment e.g., electron-phonon, electron-electron interactions.

A parameter that may be the least determined physically is the probe's position $x_0$, and how many of these probes should be placed in the tunneling/spin active region. Although it is beyond the scope of this work, one may estimate how many electron-phonon events may occur as a function of temperature and the particular electron-phonon coupling within the tunneling region and statistically place as many probes as the estimate dictates. As to the specific

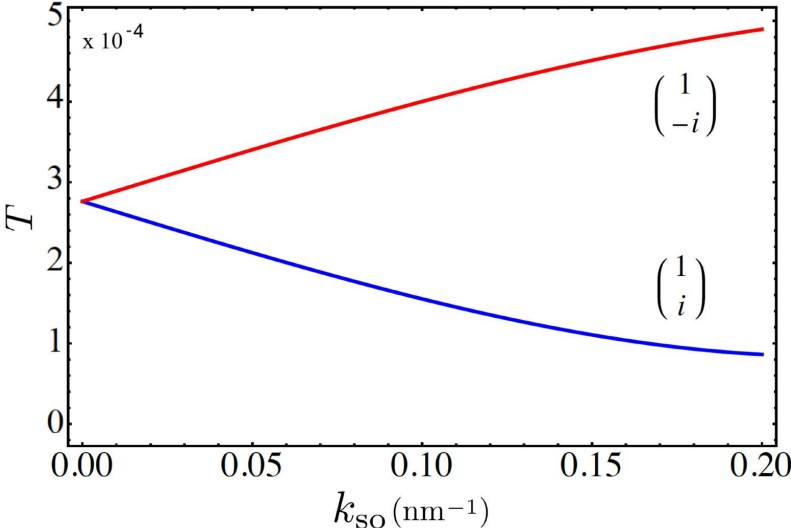

Figure 10: Transmission contrast of spin orientations along the $y$ quantization axis ($\mathbf{B}_{SO}$ field orientation). The input spin wavefunction in the (1 0) orientation, which has equal amplitudes in the latter basis, acquires a preferred orientation aligned with the $\mathbf{B}_{SO}$ field analogously to the $U(1)$ magnetic field case.

location of the probe, the most appropriate procedure is to average its effect over different positions under the barrier. We have not performed these procedures but show the smooth behavior of the polarization as a function of the probe position so that the previous procedures can be readily implemented. Figure 14 shows the smooth probe position dependence of the decoherence-generated spin polarization for the parameters of Fig. 11. The figure also displays the sensitivity of the generated polarization as a function of probe-barrier coupling $\varepsilon$. Such smooth dependence lends itself to the averaging procedures proposed and will not change qualitatively the one probe results in an essential way.

## 6 Summary and Discussion

We have discussed a one-dimensional system with SO interaction which can be derived from a three-dimensional model ignoring the orbital degree of freedom [24, 25]. The spin-orbit coupling arises from the geometrical arrangement of the $p$ orbitals of the helical model's chiral structure. Without it, the SO coupling would be orders of magnitude smaller [25], as happens comparing SO coupling of planar graphene and carbon nanotubes [1]. Due to the atomic origin of the SO coupling in this model, the helix's spin and orbital degrees of freedom are uncoupled at the lowest order and in the half-filling model [25, 26]. Thus the orbital degree of freedom only modulates the kinetic energy and adds orbital angular momentum, which can also enhance spin-orbit effects, as shown in ref. [33]. So chirality has a role in the present CISS model in generating its ∼10 meV strength.

In the succession of models presented for transmission through a spin-active barrier, we have expressed spin polarization as the manifestation of a lack of TRS that selects a spin orientation. We first discussed Büttiker's model of a $U(1)$ magnetic field under a barrier. The differential decay of the amplitudes for spin-up and spin-down produces a reorientation of the spin along the magnetic field. It serves as the trademark of TRS breaking in the spin polarization. In the following model we assessed the SO coupling where an effective magnetic field can also be identified $\mathbf{B}_{SO}$, mapping the SO coupling to a Zeeman-like term. Nevertheless, this

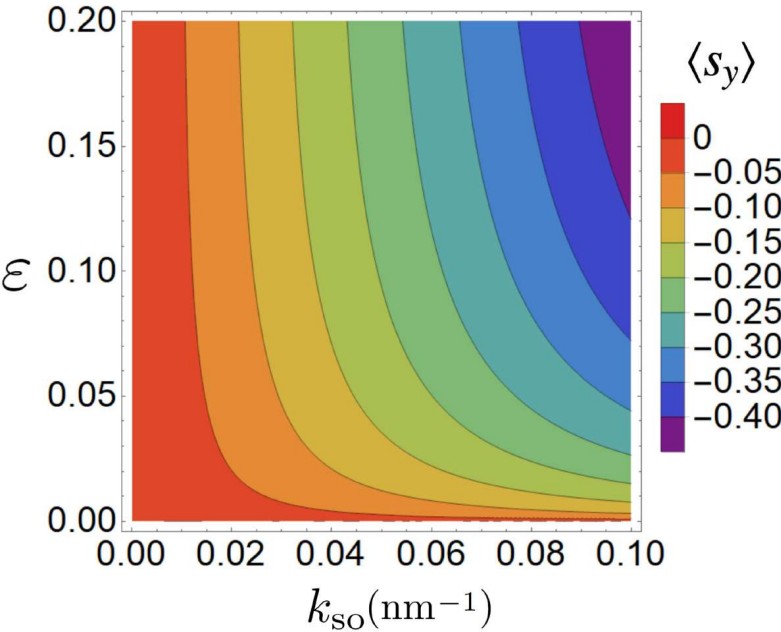

Figure 11: Spin polarization for $k = 0.4$ nm$^{-1}$, $k_0 = 1$ nm$^{-1}$, $a = 5$ nm, $x_0 = 1.5$ nm and $\mathcal{N} = 0.1$, generated by decoherence analogous to that caused by a real external magnetic field. The contour plot shows the effect of a spin-orbit and decoherence coupling, consistent with estimates of ref. [24] for tunneling. The appearance of alignment of the spin to the $\mathbf{B}_{SO}$ is very sensitive to the coupling to the Büttiker's probe, producing up to 16% polarization with realistic values of SO coupling.

effective field depends on the propagation direction and does not break TRS. The exact solution of the tunneling problem, which has not been satisfactorily solved before in the literature in this form, yields nevertheless the expected result, implied by Onsager's reciprocity for two terminal devices, i.e. *no spin polarization* independent of the magnitude of the SO coupling. Spin precession around $\mathbf{B}_{SO}$ with no differential decay of different spin components is shown. Thus, in the two terminal setup, at $T = 0$, the chiral structure and SO will then not be enough for spin selectivity since, as we have shown above, the spin-orbit coupling only makes for spin precession due to the spin torque of the SO magnetic field $\mathbf{B}_{SO}$ with no asymmetric treatment of both spin species.

One of the most emblematic features of the CISS effect is that it is measured at room temperature (although see [34,35]) in molecular systems that are strongly coupled to the thermal environment. Strictly coherent quantum models can only be part of the story. The final model addresses minimal coupling to the environment through a third probe, making for decoherent/dephasing albeit unitary processes. Symmetry arguments cannot assess beforehand, how sensitive the system will be to weak-TRS-breaking events. These events are incorporated in the last model as a time reversal symmetric interaction (SO coupling) under a barrier coupled to a Büttiker's probe. Our exact results show a high sensitivity of spin polarization, where the combination of SO coupling and decoherence acts analogously to a $U(1)$ magnetic field. The input spin reorients to the effective magnetic field producing a net spin polarization. This is a very different mechanism from the first model considered, where differential decay of each spin orientation gives polarization. Here delicate interferences that guarantee time-reversal symmetry are disrupted, producing a large effect of even 40% for small couplings to the Büttiker's probe.

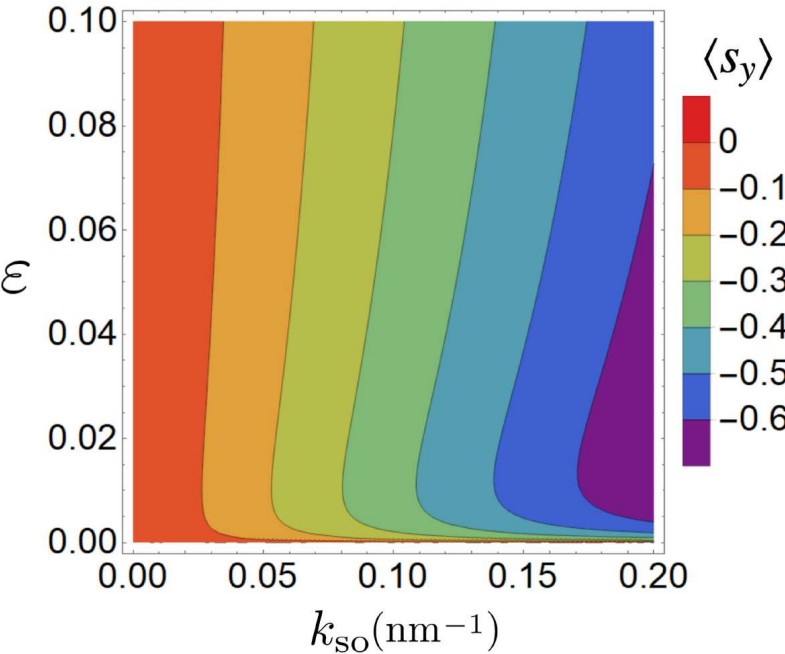

Figure 12: Spin polarized component for $k = 0.6$ nm$^{-1}$, $k_0 = 2$ nm$^{-1}$, $a = 4$ nm, $x_0 = 1.5$ nm and $\mathcal{N} = 0.1$. Manifest interference effects in the spin polarization through the coupling to the third probe. Small couplings to the probe can produce large polarizations while larger couplings degrade the polarization. Note the dependence on the probe position $x_0$ and the high sensitivity to the probe coupling that cannot be surmised solely based on symmetry arguments.

The proposed scenario, in the context of an exactly solved model, addresses many issues that have related to the CISS effect: i) The size of the SO coupling is set to realistic values in agreement with theoretical estimates in the meV range [25]; ii) The models reproduce the Onsager relations for two terminal devices for TRS interactions; iii) Coupling weakly to the environment through a third probe (other than terminals) produces highly sensitive effects of the polarization capacity of chiral molecules, that can match the order of magnitude found in experiments. Additionally, we note that other interactions such as electron- and spin-phonon have been modeled [36] to investigate their effect on electron transport and spin polarization through chiral molecules. These works show that non-zero coupling to a thermal reservoir is necessary to have spin selectivity [37–40].

Of course, this is the bare-bones model for the spintronics of chiral molecules. The nature of the environment coupling to electron transport should be developed in much more detail to assess its quantitative correctness. Incorporating the interplay between orbital and spin degrees of freedom, not yet addressed to our knowledge, should enrich further the possibilities of the theory to describe, more completely, the CISS effect.

In summary, one of our main conclusions in that decoherence effects, even those associated with small coupling constants, can translate into significant changes in spin polarization. This important result allows for a Büttiker-probe representation of several mechanisms, including electron-electron and electron-phonon interactions, that can be related to the CISS effect. There is an important connection between our treatment and a Liouville equation description of electron transfer in molecules [41], that we will explore extensively in a forthcoming article.

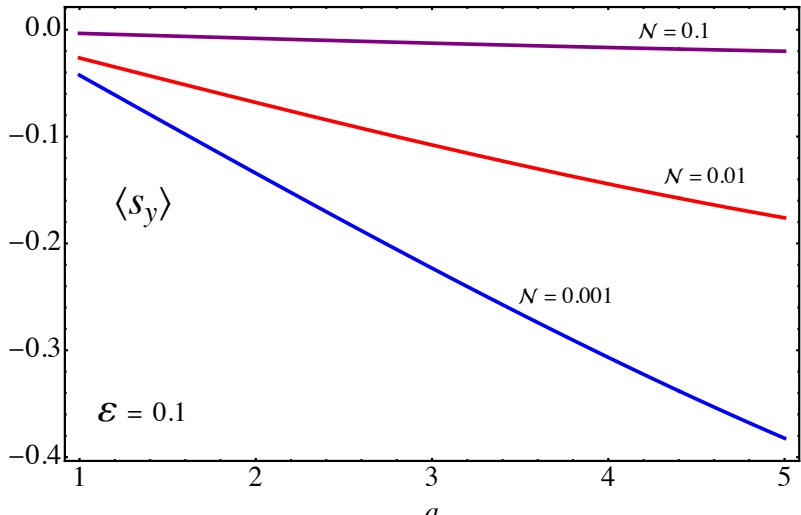

Figure 13: Temperature dependence of spin polarization as a function of the barrier length and occupation of the Büttiker's probe for $k = 0.4$ nm$^{-1}$, $k_{\text{so}} = 0.1$ nm$^{-1}$, $k_0 = 1$ nm$^{-1}$, $x_0 = 1.5$ nm and $\varepsilon = 0.1$ (same as those of Fig. 11). The polarization increases with temperature for the range of parameters chosen.

# Acknowledgements

**Funding information**   E.M. acknowledges support from Project 17617 Spin Active Molecular electronics by the Universidad San Francisco de Quito. M.P. acknowledge the support given by the Dresden Junior Fellowship Programme by the Chair of Materials Science and Nanotechnology at the Technische Universität Dresden. S.V. acknowledges the support given by the Eleonore-Trefftz-Programm at the Technische Universität Dresden. V.M acknowledges the support of Ikerbasque, the Basque Foundation for Science, and the W.M. Keck Foundation through the grant "Chirality, spin coherence and entanglement in quantum biology".

# A   Coupling Büttiker's Probe

To implement Büttiker's probe, we divide the scattering problem into four regions, two inside the barrier to accommodate for a probe at site $x_0$.

$$\psi_1 = \begin{pmatrix} \frac{1+s}{2} \\ \frac{1-s}{2} \end{pmatrix} e^{ikx} + \begin{pmatrix} A_+ \\ A_- \end{pmatrix} e^{-ikx},$$

$$\psi_2 = \frac{\alpha}{\sqrt{2}} \begin{pmatrix} 1 \\ i \end{pmatrix} e^{iq_+^+ x} + \frac{\beta}{\sqrt{2}} \begin{pmatrix} 1 \\ -i \end{pmatrix} e^{iq_-^+ x} + \frac{\gamma}{\sqrt{2}} \begin{pmatrix} 1 \\ i \end{pmatrix} e^{iq_+^- x} + \frac{\delta}{\sqrt{2}} \begin{pmatrix} 1 \\ -i \end{pmatrix} e^{iq_-^- x},$$

$$\psi_\pm^{probe} = \sqrt{\mathcal{N}} (\zeta_\pm e^{i\kappa y} + e^{-i\kappa y}) \begin{pmatrix} 1 \\ \pm i \end{pmatrix},$$

$$\psi_3 = \frac{\alpha'}{\sqrt{2}} \begin{pmatrix} 1 \\ i \end{pmatrix} e^{iq_+^+ x} + \frac{\beta'}{\sqrt{2}} \begin{pmatrix} 1 \\ -i \end{pmatrix} e^{iq_-^+ x} + \frac{\gamma'}{\sqrt{2}} \begin{pmatrix} 1 \\ i \end{pmatrix} e^{iq_+^- x} + \frac{\delta'}{\sqrt{2}} \begin{pmatrix} 1 \\ -i \end{pmatrix} e^{iq_-^- x},$$

$$\psi_4 = \begin{pmatrix} D_+ \\ D_- \end{pmatrix} e^{ikx}, \tag{A.1}$$

where now $\alpha, \beta, \gamma, \delta$, and $\alpha', \beta', \gamma', \delta'$ correspond to amplitudes at either sides of $x_0$. We assume the wideband limit for the probe leads so that the reservoir lead can carry any wavevector

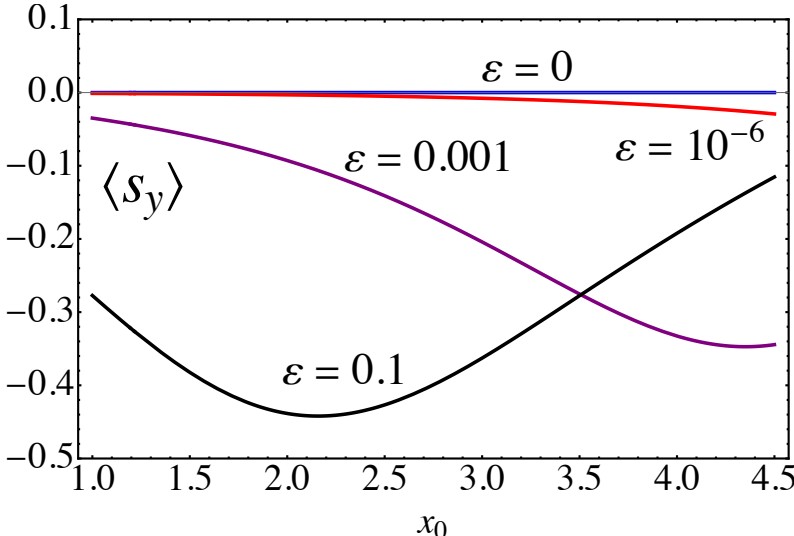

Figure 14: Dependence of the polarization on the probe position $x_0$ under the barrier, for the parameters of Fig.11, $k = 0.4\,\mathrm{nm}^{-1}$, $k_{\mathrm{so}} = 0.1\,\mathrm{nm}^{-1}$, $k_0 = 1\,\mathrm{nm}^{-1}$ for different probe barrier couplings $\varepsilon$. We see the sensitivity of the polarization to the coupling of the probe and the smooth behavior of the polarization with the probe position.

under the barrier without wavevector-dependent scattering [42]. $\mathcal{A}$, and $\mathcal{B}$ were defined below Eq.26 Such amplitude are now related by the scattering matrix

$$\begin{pmatrix} \sqrt{\mathcal{N}}\zeta_+ \\ \alpha' \\ \gamma \end{pmatrix} = \begin{pmatrix} -(\mathcal{A}+\mathcal{B}) & -\sqrt{\varepsilon}e^{iq_+^- x_0} & \sqrt{\varepsilon}e^{iq_+^+ x_0} \\ \sqrt{\varepsilon}e^{-iq_+^+ x_0} & -\mathcal{A}e^{-2i\Delta x_0} & \mathcal{B} \\ -\sqrt{\varepsilon}e^{-iq_+^- x_0} & \mathcal{B} & -\mathcal{A}e^{2i\Delta x_0} \end{pmatrix} \begin{pmatrix} \sqrt{\mathcal{N}} \\ \gamma' \\ \alpha \end{pmatrix}, \qquad (A.2)$$

and

$$\begin{pmatrix} \sqrt{\mathcal{N}}\zeta_- \\ \beta' \\ \delta \end{pmatrix} = \begin{pmatrix} -(\mathcal{A}+\mathcal{B}) & -\sqrt{\varepsilon}e^{iq_-^- x_0} & \sqrt{\varepsilon}e^{iq_-^+ x_0} \\ \sqrt{\varepsilon}e^{-iq_-^+ x_0} & -\mathcal{A}e^{-2i\Delta x_0} & \mathcal{B} \\ -\sqrt{\varepsilon}e^{-iq_-^- x_0} & \mathcal{B} & -\mathcal{A}e^{2i\Delta x_0} \end{pmatrix} \begin{pmatrix} \sqrt{\mathcal{N}} \\ \delta' \\ \beta \end{pmatrix}. \qquad (A.3)$$

The parameters were all defined below Eq.26. Each matrix equation involves and single spin orientation in a separate lead. $\mathcal{N}$ involves the reservoir parameters and is common to the two reservoir leads. The system has fourteen unknowns and fourteen conditions from which all variables can be found explicitly.

Here we give the expressions for the polarized transmission amplitudes:

$$\begin{aligned}
(De^{iak}/\Delta)t_+ = &\ \mathcal{B}^3(k-\Delta)^2\Big\{ k(1+i)(1-is)e^{i(a+x_0)(q_-^-+q_-^++q_+^+)+ix_0 q_+^-} \\
&+ k(1+i)(s-i)e^{i(a+x_0)(q_-^++q_+^-+q_+^+)+ix_0 q_-^-} \\
&+ \sqrt{2\mathcal{N}\epsilon}(k-\Delta)\Big[ e^{i(a+x_0)(q_-^-+q_+^+)+i(a q_+^-+x_0 q_-^-)} + e^{i(a+x_0)(q_-^++q_+^+)+i(a q_-^-+x_0 q_+^-)} \Big]\Big\} \\
&+ \mathcal{B}(k+\Delta)^2\Big\{ k(1+i)(i-s)e^{i(a+x_0)(q_-^-+q_+^-+q_+^+)+ix_0 q_-^+} \\
&- k(1-i)(i+s)e^{i(a+x_0)(q_-^-+q_+^-+q_+^-)+ix_0 q_+^+} \\
&- \sqrt{2\mathcal{N}\epsilon}\Big[ (k-\Delta)\Big(e^{i(a+x_0)(q_-^-+q_+^+)+i(a q_+^-+x_0 q_-^+)} + e^{i(a+x_0)(q_-^++q_+^-)+i(a q_-^-+x_0 q_+^+)}\Big)\Big]\Big\} \\
&+ \mathcal{A}^2(k-\Delta)^2\Big\{ \mathcal{B}k(1+i)(i-s)e^{i(a+x_0)(q_-^++q_+^-+q_+^+)+ix_0 q_-^-} \\
&- \mathcal{B}k(1-i)(i+s)e^{i(a+x_0)(q_-^-+q_+^-+q_+^-)+ix_0 q_+^-} \\
&- \sqrt{2\mathcal{N}\epsilon}(k-\Delta)e^{i(a+x_0)(q_-^++q_+^+)}\Big(e^{i(a q_+^-+x_0 q_-^-)} + e^{i(a q_-^-+x_0 q_+^-)}\Big)
\end{aligned}$$

$$+\sqrt{2\mathcal{N}}\epsilon(k+\Delta)\Big[e^{ia(q_-^++q_+^-+q_+^+)+ix_0(q_-^-+q_+^++q_+^-)}+e^{ia(q_+^++q_+^-+q_+^+)+ix_0(q_-^-+2q_+^-)}$$
$$e^{ia(q_-^++q_+^-+q_+^+)+ix_0(2q_-^-+q_+^+)}+e^{ia(q_-^-+q_+^-+q_+^+)+ix_0(2q_-^-+q_+^+)}$$
$$e^{ia(q_-^-+q_-^++q_+^+)+ix_0(q_-^-+q_+^-+q_+^+)}+e^{ia(q_-^-+q_-^++q_+^-)+ix_0(q_+^++2q_+^-)}\Big]\Big\}$$

$$+\mathcal{A}(k-\Delta)\Big\{\mathcal{B}k(k+\Delta)(1+i)(1-is)e^{i(a+x_0)(q_-^-+q_+^+)+i(aq_+^-+2x_0q_+^-)}$$
$$+\mathcal{B}k(k+\Delta)(1+i)(1-is)e^{i(a+x_0)(q_-^-+q_-^+)+i(aq_+^-+2x_0q_+^+)}$$
$$+k(1+i)(s-i)e^{i(a+x_0)(q_+^-+q_+^+)}\Big(e^{i(aq_-^++2x_0q_-^-)}+e^{i(aq_-^-+2x_0q_-^+)}\Big)$$
$$+\sqrt{2\mathcal{N}}\epsilon(k-\Delta)\Big[e^{ia(q_-^-+q_+^++q_+^+)+ix_0(q_+^-+2q_+^-)}+e^{ia(q_+^-+q_+^-+q_+^+)+ix_0(2q_-^-+q_+^+)}$$
$$+e^{ia(q_-^-+q_+^-+q_+^+)+ix_0(2q_+^-+q_+^+)}+e^{ia(q_-^-+q_-^++q_+^-)+ix_0(q_+^-+2q_+^+)}\Big]$$
$$-\mathcal{B}^2\sqrt{2\mathcal{N}}\epsilon(k-\Delta)^2e^{i(a+x_0)(q_+^-+q_+^+)}\Big(e^{i(aq_+^-+x_0q_-^-)}+e^{i(aq_-^-+x_0q_+^-)}\Big)$$
$$+\sqrt{2\mathcal{N}}\epsilon(k+\Delta)^2\Big[e^{ia(q_+^++q_+^-+q_+^+)+ix_0(2q_-^-+q_+^-)}+e^{ia(q_-^-+q_+^-+q_+^+)+ix_0(2q_+^-+q_+^-)}$$
$$e^{ia(q_-^-+q_+^++q_+^+)+ix_0(q_-^-+2q_+^-)}+e^{ia(q_-^-+q_+^-+q_+^+)+ix_0(q_-^-+q_+^++q_+^+)}$$
$$e^{ia(q_-^-+q_+^++q_+^-)+ix_0(q_+^-+q_+^-+q_+^+)}+e^{ia(q_-^-+q_+^++q_+^-)+ix_0(q_-^-+2q_+^-)}\Big]$$
$$+\mathcal{A}^3\sqrt{2\mathcal{N}}\epsilon(k-\Delta)^3e^{i(a+x_0)(q_+^-+q_+^+)}\Big(e^{i(aq_+^-+x_0q_-^-)}+e^{i(aq_-^-+x_0q_+^-)}\Big)$$
$$+\mathcal{B}^2\sqrt{2\mathcal{N}}\epsilon(k-\Delta)^2(k+\Delta)\Big[e^{i(a+x_0)(q_+^-+q_+^-)+i(aq_+^-+x_0q_-^-)}$$
$$+e^{i(a+x_0)(q_-^-+q_+^+)+i(aq_-^-+x_0q_+^-)}\Big]$$
$$-\sqrt{2\mathcal{N}}\epsilon(k+\Delta)^3e^{i(a+x_0)(q_-^-+q_+^-)}\Big[e^{i(aq_+^-+x_0q_-^+)}+e^{i(aq_-^-+x_0q_+^+)}\Big]\Big\}, \tag{A.4}$$

$$(De^{iak}/k)t_- = \mathcal{B}(k+\Delta)^2\Big\{\Delta(1-i)(s-i)e^{i(a+x_0)(q_-^-+q_+^-+q_+^+)+ix_0q_-^+}$$
$$+\Delta(1+i)(i+s)e^{i(a+x_0)(q_-^-+q_+^++q_-^-)+ix_0q_+^+}$$
$$-i\sqrt{2\mathcal{N}}\epsilon(k-\Delta)\Big[e^{i(a+x_0)(q_-^-+q_+^+)+i(aq_+^-+x_0q_-^+)}+e^{i(a+x_0)(q_+^-+q_+^+)+i(aq_-^-+x_0q_+^+)}\Big]\Big\}$$
$$+\mathcal{B}^3(k-\Delta)^2\Big\{\Delta(1+i)(1+is)e^{i(a+x_0)(q_+^-+q_+^-+q_+^+)+ix_0q_-^-}$$
$$-\Delta(1+i)(i+s)e^{i(a+x_0)(q_-^-+q_+^++q_+^+)+ix_0q_+^-}$$
$$+i\sqrt{2\mathcal{N}}\epsilon(k-\Delta)\Big[e^{i(a+x_0)(q_+^-+q_+^+)+i(aq_+^-+x_0q_-^-)}-e^{i(a+x_0)(q_+^-+q_+^+)+i(aq_-^-+x_0q_+^-)}\Big]\Big\}$$
$$+\mathcal{A}^2(k-\Delta)^2\Big\{\mathcal{B}\Delta(1-i)(s-i)e^{i(a+x_0)(q_-^-+q_+^-+q_+^+)+ix_0q_-^-}$$
$$+\mathcal{B}\Delta(1+i)(i+s)e^{i(a+x_0)(q_-^-+q_+^++q_+^+)+ix_0q_+^-}$$
$$-i\mathcal{B}\sqrt{2\mathcal{N}}\epsilon(k-\Delta)e^{i(a+x_0)(q_+^-+q_+^+)}(e^{i(aq_+^-+x_0q_-^-)}-e^{i(aq_-^-+x_0q_+^-)})$$
$$-i\sqrt{2\mathcal{N}}\epsilon(k+\Delta)\Big[e^{ia(q_+^-+q_+^-+q_+^+)+ix_0(q_-^-+q_+^++q_+^-)}-e^{ia(q_-^-+q_+^-+q_+^+)+ix_0(q_+^-+2q_+^-)}$$
$$+e^{ia(q_+^-+q_+^-+q_+^+)+ix_0(2q_-^-+q_+^+)}+e^{ia(q_-^-+q_+^-+q_+^+)+ix_0(2q_+^-+q_+^+)}$$
$$-e^{ia(q_-^-+q_+^++q_+^+)+ix_0(q_-^-+q_+^-+q_+^+)}-e^{ia(q_-^-+q_+^++q_+^-)+ix_0(q_+^-+2q_+^+)}\Big]\Big\}$$
$$+\mathcal{A}(k-\Delta)\Big\{\mathcal{B}\Delta(k+\Delta)(1+i)(1+is)e^{i(a+x_0)(q_-^-+q_+^+)}\Big[e^{i(aq_+^-+2x_0q_-^-)}+e^{i(aq_-^-+2x_0q_+^-)}\Big]$$
$$-\mathcal{B}\Delta(k+\Delta)(1+i)(i+s)e^{i(a+x_0)(q_-^-+q_+^-)}\Big[e^{i(aq_+^-+2x_0q_-^-)}+e^{i(aq_+^-+2x_0q_+^-)}\Big]$$
$$+i\mathcal{B}(k^2-\Delta^2)\sqrt{2\mathcal{N}}\epsilon\Big[-e^{ia(q_-^-+q_+^-+q_+^+)+ix_0(q_+^-+2q_+^-)}+e^{ia(q_+^-+q_+^-+q_+^+)+ix_0(2q_-^-+q_+^+)}$$
$$+e^{ia(q_-^-+q_+^-+q_+^+)+ix_0(2q_+^-+q_+^+)}-e^{ia(q_-^-+q_-^++q_+^-)+ix_0(q_+^-+2q_+^+)}\Big]$$

$$-i\mathcal{B}^2(k-\Delta)^2\sqrt{2\mathcal{N}}\epsilon\Big[e^{ia(q_-^++q_+^-+q_+^+)+ix_0(q_-^-+q_-^++q_+^+)} - e^{ia(q_-^-+q_-^++q_+^+)+ix_0(q_-^++q_+^-+q_+^+)}\Big]$$

$$+i(k+\Delta)^2\sqrt{2\mathcal{N}}\epsilon\Big[e^{ia(q_-^++q_+^-+q_+^+)+ix_0(2q_-^-+q_+^-)} + e^{ia(q_-^-+q_+^-+q_+^+)+ix_0(2q_-^++q_+^-)}$$

$$-e^{ia(q_-^-+q_-^++q_+^+)+ix_0(q_-^-+2q_-^+)} + e^{ia(q_-^-+q_+^-+q_+^+)+ix_0(q_-^-+q_-^++q_+^+)}$$

$$-e^{ia(q_-^-+q_-^++q_+^-)+ix_0(q_+^++q_+^-+q_+^+)} - e^{ia(q_-^-+q_-^++q_+^-)+ix_0(q_-^-+2q_+^+)}\Big]$$

$$+i\mathcal{A}^3(k-\Delta)^3\sqrt{2\mathcal{N}}\epsilon\Big[e^{ia(q_-^++q_+^-+q_+^+)+ix_0(q_-^-+q_-^++q_+^+)} - e^{ia(q_-^-+q_-^++q_+^+)+ix_0(q_-^++q_+^-+q_+^+)}\Big]$$

$$+i\mathcal{B}^2(k-\Delta)^2(k+\Delta)\sqrt{2\mathcal{N}}\epsilon\Big[e^{ia(q_-^++q_+^-+q_+^+)+ix_0(q_-^-+q_-^++q_+^-)}$$

$$-e^{ia(q_-^-+q_-^++q_+^+)+ix_0(q_-^-+q_+^-+q_+^+)}\Big]$$

$$-i(k+\Delta)^3\sqrt{2\mathcal{N}}\epsilon\Big[que^{ia(q_-^-+q_+^-+q_+^+)+ix_0(q_-^-+q_-^++q_+^-)}$$

$$-e^{ia(q_-^-+q_+^-+q_+^-)+ix_0(q_-^-+q_-^++q_+^+)}\Big]\Big\}, \tag{A.5}$$

$$D =\Big[-\mathcal{A}^2(k-\Delta)^2\ e^{i\left(aq_-^++x_0(q_-^-+q_-^+)\right)} + \mathcal{A}(k^2-\Delta^2)\Big(e^{i(aq_-^++2x_0q_-^-)} + e^{i(aq_-^-+2x_0q_-^+)}\Big)$$

$$+e^{ix_0(q_-^-+q_-^+)}\big(\mathcal{B}^2(k-\Delta)^2e^{iaq_-^+} - (k+\Delta)^2e^{iaq_-^-}\big)\Big]$$

$$\times\Big[\mathcal{A}^2(k-\Delta)^2\ e^{i\left(aq_+^++x_0(q_+^-+q_+^+)\right)} - \mathcal{A}(k^2-\Delta^2)\Big(e^{i(aq_+^++2x_0q_+^-)} + e^{i(aq_+^-+2x_0q_+^+)}\Big)$$

$$+e^{ix_0(q_+^-+q_+^+)}\Big(-\mathcal{B}^2(k-\Delta)^2e^{iaq_+^+} + (k+\Delta)^2e^{iaq_+^-}\Big)\Big]. \tag{A.6}$$

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
