# Peer review of "Spin polarization induced by decoherence in a tunneling one-dimensional Rashba model"

_SciPost Physics Core, doi:SciPost Phys. Core 6, 044 (2023)_

## Round 1 · Referee Report · Viktor Könye (Referee 1) · 2023-2-9

Strengths

1-, Very intuitive toy models to understand the phenomenon.

2-, Clear easy to follow didactic explanation of the previously known results.

Weaknesses

1-, The new result involving the Buttiker's probe is not detailed enough and hard to reproduce.

2-, Not very clear Figures and captions, with parameters missing many times.

3-, Sometimes hard to distinguish what is new and what has been known in the literature. Which also shadows the importance of the result.

Report

The Authors study simple one-dimensional effective models to qualitatively explain the Chiral-Induced Spin Selectivity (CISS) effect. They show what are the minimal ingredients needed in a 1d scattering setup to achieve such an effect.

I find the final result interesting and an important step towards a deeper understanding of the CISS effect.

The Authors start from a very didactic review of previously known results that is very accessible even to those not in the field. But then in Sec 5 which is the main result section of the paper the formalism and results are much more obscure (see comments for more details on this).

It is sometimes hard to distinguish what is an actual new result achieved by the Authors and what is already well known (more on this in the comments too). The way the manuscript reads for me is that there are all these results and methods already well known and this is a short review on those and finally the methods are all combined to qualitatively explain something new.

Because of the points mentioned above in my opinion the way the paper is presented, as of now, is more suitable for Scipost Physics Core. I also think that before being accepted in any journal there are several points that should be addressed.

Comments and questions:

1,- The Authors mention quantitative manner in the abstract which is a bit misleading in my opinion. The results seem qualitative to me.

2-, From what I understood Sections 2,3,4 do not really contain new results they are mainly rewriting Refs [18-20] in a more coherent format. If this is the case then I would avoid statements as 'we exactly solve' , 'we show' and rather indicate that these results are summaries from the literature. If this is not the case and these sections do include new previously unpublished and not understood results then they should be better highlighted.

3-, Section 5 contains the main result of the paper. However in my opinion this section is the least cleanly written in terms of formulas. While reproducing the results in Sections 2-4 is very straightforward, the ones of Sec 5 are quite obscure. Maybe the formulas are very complicated and don't fit in the main paper but they should at least appear in an appendix. Several things are not defined such as $\Gamma$ or $\kappa$ in the Fig 8. I would have expected that all the definitions and methodical explanations in Secs 2-4 are there so they can be refered to in Sec 5, but I saw no reference to any of the previous equations. Even though plenty of the variables in Sec 5 have been defined previously. This makes it very hard to connect this section to the previous ones.

4-, what does $t_{total}$ mean without the square? Not sure why is it necessary to introduce that quantity.

5-, The meaning of $x_0$ is not really clear for me. How does one choose this value? Why is it 1.5nm in Fig 12 but 0.8nm in Fig 13?

6-, A few questions about Figure 9. If I understood it correctly, the whole point is the appearance of $<s_y>$ but that is not shown on the plot. Also how is it possible that at 0 width the state does not start from (1 0) according to the graph? Is that the effect of the decoherence probe? But then shouldn't it vanish somehow in the limit of no scattering region?

7-, Figures 10-13 are a bit confusing. Parameters are explicitly shown for Figs. 12-13 but not in other Figures. What is the difference between Figure 11 and Figure 12? Why is Figure 12 and 13 for different parameters? How are these parameters chosen?

Requested changes

1-, After Eq (9) $v_x=\partial H/\partial x /m$ seems like a typo or otherwise I don't understand the formula. The definiton after Eq. (16) seems to be the correct one.

2-, Eq. (21) is the same as Eqs. (8) (9) not sure why they need to be written twice.

3-, In Eq (22) t and r just appear without definition. One would assume that the definitons are the same as in Eq. (10) but since the normalization of the wave functions are different I am not sure this is the case. I would suggest to remove A and D completely and just use r and t. Or redefine r and t each time.

4-, Figure 11 and 12 do not have a label for the colorbar. Colorscale could be linear in greyscale and with better resolution.

  • validity: good
  • significance: good
  • originality: ok
  • clarity: low
  • formatting: reasonable
  • grammar: reasonable

Author:  Bertrand Berche  on 2023-02-17  [id 3365]

(in reply to Report 1 by Viktor Könye on 2023-02-09)
Category:
answer to question

Comments and questions:

REFEREE
1,- The Authors mention quantitative manner in the abstract which is a bit misleading in my opinion. The results seem qualitative to me.

RESPONSE
In the First three sections are indeed qualitative, as the parameters are not chosen to fit the actual physical ones for chiral molecules. In section 3, nevertheless, independently of the parameters chosen, the null results for polarization are exact results verifying symmetry requirements on the Onsager relations. Many previous numerical computations rendered transport filtering associated with non-hermiticities that crept into the calculation. Although we cite previous papers with the correct conclusion for the polarization of the Rashba case, these contributions have shortcomings that are overcome here realizing the correct boundary conditions.

The meaning of polarization in terms of an asymmetric treatment of opposing spin orientations is also clarified in this paper, along with an exact solution to the problem (with TRS), in light of Buttiker’s magnetic field case (breaking TRS).

In section 5, on Buttiker’s probe, the parameters are fitted to a polaron model of transport, and quantitative values for polarization apply to chiral molecules. In this sense, the model is quantitative.

REFEREE
2-, From what I understood Sections 2,3,4 do not really contain new results they are mainly rewriting Refs [18-20] in a more coherent format. If this is the case then I would avoid statements as 'we exactly solve' , 'we show' and rather indicate that these results are summaries from the literature. If this is not the case and these sections do include new previously unpublished and not understood results then they should be better highlighted.

RESPONSE
This point has been clarified in the previous response. The only rewriting that we have performed as a reference problem is that of the tunneling problem in the presence of a magnetic field (Buttiker’s case). The results for the Rashba tunneling are, as far as we know, the first consistent solution to the problem, although we reach the expected conclusions from the corresponding symmetry arguments.

REFEREE
3-, Section 5 contains the main result of the paper. However in my opinion this section is the least cleanly written in terms of formulas. While reproducing the results in Sections 2-4 is very straightforward, the ones of Sec 5 are quite obscure. Maybe the formulas are very complicated and don't fit in the main paper but they should at least appear in an appendix. Several things are not defined such as
Γ or κ in the Fig 8. I would have expected that all the definitions and methodical explanations in Secs 2-4 are there so they can be referred to in Sec 5, but I saw no reference to any of the previous equations. Even though plenty of the variables in Sec 5 have been defined previously. This makes it very hard to connect this section to the previous ones.

RESPONSE
We agree with the referee to expand in more detail section 5, including an appendix, connecting with equations of sections 2-4. Some of the significant expressions involved can then be checked for correctness by the readers. We will clarify all missing definitions mentioned by the referee and revise the document for further omissions.

REFEREE
4-, what does t_total mean without the square? Not sure why is it necessary to introduce that quantity.

RESPONSE
We agree with the referee, we will eliminate this new definition which is not meaningful and is not used anywhere else in the paper.

REFEREE
5-, The meaning of x0 is not really clear for me. How does one choose this value? Why is it 1.5nm in Fig 12 but 0.8nm in Fig 13?

RESPONSE
Buttiker’s voltage probe is a local device and must be introduced at a particular position under the barrier. There is a small but noticeable effect on the final polarization result. What is intended in the model is that a single decoherent event is included. This x0 might be the site of an electron-phonon interaction where the phonons are thermalized. In a single molecule, tunneling processes may include a number of these events depending on the nature of the interaction coupling to the reservoir.

The value for x0 was chosen in figure 12 to illustrate the reentrant regime for the polarization (it can be non-monotone as in figure 11) this was most easily accessed by modifying the voltage probe point. The x0 chosen for Figure 13 is the same as that of figure 12. We will clarify this in the new version of the article.

REFEREE
6-, A few questions about Figure 9. If I understood it correctly, the whole point is the appearance of <sy> but that is not shown on the plot. Also how is it possible that at 0 width the state does not start from (1 0) according to the graph? Is that the effect of the decoherence probe? But then, shouldn't it vanish somehow in the limit of no scattering region?

RESPONSE
We thank the referee for having caught this mistake, in fact, the figure should not have started from a=0 since the voltage probe is set to be at x0=0.8 in this plot. That means the voltage probe is outside the barrier. We will introduce a correct figure departing from a value of ‘a’ the barrier length starts beyond x0. There the polarization cannot begin at the entrance polarization since they evolve to the output of a barrier of length a.

In fact we were careful in the paper, for the polarization plots, that we did not make this mistake. We will clarify all parameters to make this clear.

REFEREE
7-, Figures 10-13 are a bit confusing. Parameters are explicitly shown for Figs. 12-13 but not in other Figures. What is the difference between Figure 11 and Figure 12? Why is Figure 12 and 13 for different parameters? How are these parameters chosen?

RESPONSE
This confusion explains in more detail how time-reversal symmetry breaking operates in this model: For the magnetic field under the barrier, the spin relaxes montonely toward the magnetic field direction. When TRS is broken by decoherence we wanted to show this relaxation is not monotone as it appears in Figure 11. Varying the point at which the voltage probe is applied shows a non-monotone evolution of the magnetization, first relaxing toward Bso then moving away as the coupling to the probe is increased. In fact this may show a route to validation in experiments. We will discuss this in the new version of the manuscript.

REFEREE
Requested changes
1-, After Eq (9) vx=∂H/∂x/m
seems like a typo or otherwise I don't understand the formula. The definition after Eq. (16) seems to be the correct one.

RESPONSE
Indeed, this is a typo. We will correct it in the new manuscript.

REFEREE
2-, Eq. (21) is the same as Eqs. (8) (9) not sure why they need to be written twice.

RESPONSE
Will refer to the first mention of these equations.

REFEREE
3-, In Eq (22) t and r just appear without definition. One would assume that the definitions are the same as in Eq. (10) but since the normalization of the wave functions are different I am not sure this is the case. I would suggest to remove A and D completely and just use r and t. Or redefine r and t each time.

RESPONSE
Indeed the definition changes somewhat because of the norm. We will make the corresponding changes defining t and r.

REFEREE
4-, Figure 11 and 12 do not have a label for the colorbar. Colorscale could be linear in greyscale and with better resolution.

RESPONSE
We will put the required label in the colorbar.

—------------------------------------

Additional Note:

We would like to acknowledge an observation made to us privately by K. Huisman: We wrongly claim in our work that his paper J. Phys. Chem. C 125, 23364 (2021), uses a non-unitary version of the Buttiker probe. In fact this paper uses the Buttikers probe with no loss of unitarity and only induces decoherence. We will amend this claim in the new version of the manuscript.

The authors

---

## Round 1 · Referee Report · Daniel Varjas (Referee 2) · 2023-2-17

Strengths

1) Application of Buttiker's decoherence probe to the CISS effect 2) Pedagogical introduction of the models and methods

Weaknesses

1) Hard to follow the relation to more realistic models and experimental results 2) Not always clear which results are new 3) Some plots are hard to read

Report

The manuscript investigates simple models of electrons tunnelling in the presence of magnetic field or spin-orbit-coupling (SOC). Beyond pedagogically introducing these models, the manuscript also studies a "Buttiker's decoherence probe" setup for SOC tunnelling, in order to explain the experimentally observed Chiral-Induced Spin Selectivity effect. The results are scientifically sound, and the presentation is overall good. I do not believe, however, that the level of novelty meets the acceptance criteria for Scipost Physics, or at the very least, the presentation should be improved to make the novel aspects of this work more emphasized. I would recommend publication in Physics Core after minor revisions (see requested changes).

Requested changes

1) The Introduction is at places hard to follow, as it skips over introducing concepts that non-experts may not take for granted. Specifically, the arguments about the first vs. second neighbour nature of SOC in the first paragraph is hard to place without any introduction of a tight-binding framework. In the first sentence of the second paragraph it is unclear what 40% refers to. Similar, hard to follow arguments relating the results to various specific systems reappear in the first paragraph of the Discussion. I ask the authors to expand on these points, so that non-specialist readers can also appreciate the results.

2) After eqn 1 “This Hamiltonian” should be clarified to mean the Hamiltonian inside the barrier.

3) Before eqn 2 “input wavefunction” should rather be incoming/incident wavefunction to keep standard terminology.

4) After eqn 9 I find the definition of the velocity operator confusing, possibly a typo. It is typically defined through the commutator of the Hamiltonian with the position operator, or its derivative with respect to momentum. The authors, however, clearly use the correct definition in the rest of the manuscript.

5) What are the parameters in Fig 4? Why is the transmission probability in the 10^-4 range?

6) The explanation about the meaning of spinors after eqn 14 feels out of place, it should either appear earlier after eqn 1, or can be omitted.

7) For Figures 11 and 12 the authors should use a continuous, perceptually uniform color scale. The current plots give the false impression of quantised values, while exceptionally difficult to read for colour-blind persons.

8) The Author contribution section should be filled out.

---

## Round 2 · Referee Report · Viktor Könye (Referee 1) · 2023-3-10

Report
Some of the problems with the original version of the manuscript were addressed. However, I still have some comments that have not been dealt with completely.
1-, About the quantitativeness of the result. As far as I understand $x_0$ and $\varepsilon$ are parameters that don't have any physically motivated values. Tuning them to reproduce the same value for the spin polarization as expected I wouldn't consider a quantitative result. For a quantitative result I would want to see what is expected and how the results compare to it. In what way do the results match with the estimates of Ref [21] or with experiments?
At the very end of the Discussion the Authors mention quantitativeness and how a more detailed study would be necessary. I think that is a more accurate stance and the statement in the abstract is too strong.
2-, Thank you for the clarification about the novelty of the results. But since the other Referee also found this unclear I think it should be more clearly stated in the manuscript as well.
3-, I am still confused about the $x_0$ dependence. This seems like an arbitrary choice to me without any physical insight. Maybe showing how results depend on $x_0$ could be beneficial. Why is $x_0$ just not fixed in the middle of the scattering region? Somehow that would feel like a straightforward choice.
4-, Figure 12 and 13 still have different parameters, at least according to the figure labels. I would suggest to put all the parameters used in every figure for full clarity and reproducibility. Or in cases where it is the same as other cases clearly mention it.
5-, It is still not clear what is the difference between Fig 11 and Fig 12 in the manuscript. Again listing all parameters would help with clarity. If I understood correctly from the response the only difference is the value of $x_0$. Which goes back to my previous question on how can one select a realistic $x_0$ value.
Author: Bertrand Berche on 2023-03-15 [id 3482]
(in reply to Report 1 by Viktor Könye on 2023-03-10)
REFEREE The manuscript improved significantly from the first submission. Some of the problems with the original version of the manuscript were addressed. However, I still have some comments that have not been dealt with completely.
1- About the quantitativeness of the result. As far as I understand $x_0$ and $\varepsilon$ are parameters that don't have any physically motivated values. Tuning them to reproduce the same value for the spin polarization as expected I wouldn't consider a quantitative result. For a quantitative result I would want to see what is expected and how the results compare to it. In what way do the results match with the estimates of Ref [21] or with experiments? At the very end of the Discussion the Authors mention quantitativeness and how a more detailed study would be necessary. I think that is a more accurate stance and the statement in the abstract is too strong.
RESPONSE The referee is correct, the parameterization of coupling parameter would depend on e.g. the magnitude of the electron-phonon or electron-electron interaction which actually talk to the thermal reservoir. We will shrink from the claim to the quantitativeness of the results in the coupling to the reservoir. This goes also for the value of $x_0$, which is part of a minimalistic coupling (a single decoherent event). We will show the dependence on this parameter under the barrier to show it has a smooth effect. Probably the best course to fit this parameter is to assess the electron-phonon scattering length to estimate how many of these events will occur under the barrier and use as many probes as are required.
The quantitativeness goes as far as the wave vectors involved in the tunneling correspond to parameters of polaron tunneling as pointed out in Ref. [21].
REFEREE 2- Thank you for the clarification about the novelty of the results. But since the other Referee also found this unclear I think it should be more clearly stated in the manuscript as well.
RESPONSE We will modify the manuscript accordingly
REFEREE 3- I am still confused about the $x_0$ dependence. This seems like an arbitrary choice to me without any physical insight. Maybe showing how results depend on $x_0$ could be beneficial. Why is $x_0$ just not fixed in the middle of the scattering region? Somehow that would feel like a straightforward choice.
RESPONSE We will add a plot of the $x_0$ dependence of the polarization. As we answered before, where the decoherence event occurs depends on how to electron transport is coupled to the reservoir via e.g. electron-phonon coupling. In the context of another probe analogous to the Buttiker probe (the D’Amato Pastawski probe), it is placed at every point along a discrete chain describing the decoherence region.
REFEREE 4- Figure 12 and 13 still have different parameters, at least according to the figure labels. I would suggest to put all the parameters used in every figure for full clarity and reproducibility. Or in cases where it is the same as other cases clearly mention it.
RESPONSE Will correct parameters used in the figure 12 and 13.
REFEREE 5- It is still not clear what is the difference between Fig 11 and Fig 12 in the manuscript. Again listing all parameters would help with clarity. If I understood correctly from the response the only difference is the value of $x_0$. Which goes back to my previous question on how can one select a realistic $x_0$ value.
RESPONSE We replied to this concern above.

Evgeny Sherman on 2023-03-06 [id 3444]
An interesting work, still, as one of the Referees, Dr. Könye, pointed out, a toy model, distant from the real physics. The authors consider only unrealistic model of spatially uniform Rashba coupling. However, in any realistic system (independent of dimensionality) it is a random function of coordinate. Direct experimental observation has been presented here:
https://www.nature.com/articles/nphys3774
Theoretical analysis shows that the effect is generic:
https://www.sciencedirect.com/science/article/abs/pii/S1386947710002213
This randomness causes dephasing of electron' spins even for straightforward motion. Taking this effect into account could possibly make the model considered by the authors realistic.
Anonymous on 2023-05-04 [id 3643]
(in reply to Evgeny Sherman on 2023-03-06 [id 3444])Indeed a periodic model is considered with no spatial inhomogeneities. This type of model can be derived from microscopic considerations e.g. tight binding, or from group theory in strictly periodic systems (See Winkler's book on SO in semiconductors). The parameters that are attempted to fit realistic values are the SO coupling the k vectors injected in a polaron type situation and the barrier height. As was mentioned by the referee's the coupling to the environment has not microscopic model associated.
Thank you for the references

---

## Round 2 · Referee Report · Daniel Varjas (Referee 2) · 2023-3-21

Report

---

## Round 2 · List of Changes

We have implemented in version 2 all the modifications suggested/recommended by the two referees.
In particular:
List of changes:
1.- Include an appendix connecting section 5 with equations of sections 2-4
2.- Clarify all missing definitions (\Gamma and k in Fig. 8). Revise the document for further omissions
3.- Eliminate |t_total| defined between equations (25) and (26)
4.- Clarify that the x_0 chosen for Fig. 13 is the same as that of Fig. 12
5.- Correct Fig. 9 with the x axis (a) starting beyond x_0 and making this clear in the text
6.- Discuss in more detail how time-reversal symmetry breaking operates in the model (Figs. 10-13)
7.- Correct typo after Eq. (9) vx=dH/dx/m
8.- Do not repeat Eq. (8) (9) in Eq. (21). Write is once and refer to it after
9.- In Eq. (22) make the corresponding changes defining t and r
10.- Put the label in the colorbar of Figs. 11 and 12
The authors

---

## Round 3 · Referee Report · Daniel Varjas (Referee 2) · 2023-5-4

Report

The authors have sufficiently addressed my previous comments, I recommend publication of the manuscript.

---

## Round 3 · List of Changes

First, we want to remind the editor/referees that for clarity, we have a pdf version of our paper with first corrections (of the first submission) highlighted in blue color and further corrections (for v2) in magenta color. This version is available upon request at bertrand.berche@univ-lorraine.fr

The first referee; Dr Konye, is correct concerning the parameterization of coupling parameter which would depend on e.g. the magnitude of the electron-phonon or electron-electron interaction which actually talk to the thermal reservoir. We shrink from the claim to the quantitativeness of the results in the coupling to the reservoir. This goes also for the value of x0, which is part of a minimalistic coupling (a single decoherent event). We now show the dependence on this parameter under the barrier to show it has a smooth effect. Probably the best course to fit this parameter is to assess the electron-phonon scattering length to estimate how many of these events will occur under the barrier and use as many probes as are required.

The quantitativeness goes as far as the wave vectors involved in the tunneling corresponding to the parameters of polaron tunneling as pointed out in Ref. [21].

We have modified the manuscript concerning the novelty of the results.

We have added a plot of the x0 dependence of the polarization. As we answered before, where the decoherence event occurs depends on how electron transport is coupled to the reservoir via e.g. electron-phonon coupling. In the context of another probe analogous to the Buttiker probe (the D’Amato Pastawski probe), it is placed at every point along a discrete chain describing the decoherence region.

The parameters used in the figure 12 and 13 are now corrected.

Concerning the queries of referee 2, Dr. Varjas, we have indicated in the introduction that the first and second neighbors refer to a tight-binding model and we have specified what the 40% polarization refers to.

The specifications that the dispersion relation refers to the Hamiltonian inside the barrier, and the change ``input’’ to ``incoming’' wave function have been done.

The velocity operator was already corrected previously.

The parameters of Fig 4 were specified (in the caption) to be the same as in Fig 3.

We have moved the introduction of the spinor definition after eqn (1), as requested by referee 2.

Color scales in Figs 11 and 12 were not modified. We believe that in a continuos color scheme, you cannot
see very well the sensitive dependence on the parameters since the eye is not so able to distinguish color changes.
The concept of level curves is well known and cannot in our opinion be misinterpreted as quantized.

Eventually, we have added the references given in the comment by Sherman.

---

## Editorial Decision

published